# Optimal Design of an Inverter-Fed PMSM for a Brake System Considering Variation in Motor Control Parameters and Input Voltage

**Kyu-Yun Hwang** [1] and **Keun-Young Yoon** [2,*]

1  2Brake R&D Center, Mechanical Hardware 1-1, Mando, Seongnam 13486, Korea; kyuyun7@hanmail.net
2  Department of Electrical Engineering, Honam University, 417 Eodeung-daero, Gwangju 62399, Korea
*  Correspondence: ky.yoon@honam.ac.kr; Tel.: +82-062-940-5408

**Abstract:** This study proposes an optimal design approach for an inverter-fed permanent magnet synchronous motor (PMSM) considering the variation in motor control parameters and input voltage (inverter output voltage), which vary with respect to the temperature and loading conditions, in an integrated brake system. In an integrated brake system, a quick response to load changes is crucial. Therefore, in this study, to consider the fluctuation in control variables and input voltage, the motor control parameters and input voltage were first calculated according to the operating temperature and loading condition. Subsequently, based on the calculated conditions and the approximated motor input voltage and control parameters, the objective functions corresponding to motor characteristics in transient and steady states were formulated using design-of-experiment (DOE), the moving least square method (MLSM), and the finite element method (FEM). Finally, the optimal design was performed using the genetic algorithm (GA). The validity of the proposed optimal design approach was verified by comparing its optimization results with the FEM analysis results. Thus, it was confirmed that a motor can be designed for an integrated brake system by considering the motor control parameters and motor input voltage, which vary with the driving conditions.

**Keywords:** integrated brake system; permanent magnet synchronous motor; optimal design; torque ripple; genetic algorithm

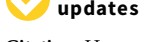



## 1. Introduction

Recently, with the changing automobile structure, electric devices have diversified, and their importance is increasing [1–5]. With the diversification of the quantity and specifications of the motors used in automobiles, the demand for motors that have been customized for various systems is also increasing. Accordingly, research is being conducted on the associated system performance analysis, considering not only the motor but also all the system components. Various co-simulations have been performed to ensure that the designed motor performs optimally in the applied system. An analysis method that combines electromagnetic and thermal analyses for the accurate analysis of heat-generation characteristics of a motor [6–8] and various co-simulation methods that combine the optimization technique with the motor design [9–11] have been reported. In addition, combined research on motor design and motor control to obtain a detailed analysis of a motor's behavior has received considerable attention. For this purpose, a recent comprehensive method [12–15] presented an improved motor control parameter modeling that considers motor performance in combination with the optimal design and a new combined optimization of the steady-state and transient-state operation of the overall system, respectively. However, these methods do not consider the variation in motor control parameters and input voltage, which vary with the ambient operating temperature and loading condition. For accurate motor performance analysis, consideration of not only the motor control parameters but also the available motor input voltage, which depend on the operating state of the motor, is

important. This is because, in general, the output power characteristics of a motor greatly vary with the variation in the motor control parameters and input voltage [16].

In this study, an optimal design process is proposed for an integrated brake system, considering the variation in motor control parameters and motor input voltage, which vary with the ambient operating temperature and loading condition. The motor control parameters and motor input voltage were calculated using the finite element method (FEM). Subsequently, the objective functions related to the motor characteristics in transient and steady states were formulated using design-of-experiment (DOE) [17], the moving least square method (MLSM) [18], and the FEM. Finally, the optimal solution of the multi-objective functions was obtained using the genetic algorithm (GA).

## 2. Optimal Design Strategy for a BLAC Motor in Integrated Brake Systems

A vehicle's integrated brake system controls its acceleration, deceleration, and stopping [19]. In particular, the integrated brake system is essential for driver safety and accident prevention because it controls the braking system in response to a collision or changes in the external environment while driving. The integrated brake system must provide rapid braking performance, regardless of changes in the external environment. In addition, as the demand for noiseless operation of a vehicle has increased in recent years, the demand in the emotional domain regarding the noise of the braking device has also greatly increased.

As the main operating source in an integrated brake system, the motor must have a fast response and low torque ripple in transient and steady-state conditions. To improve the motor performance, we developed an optimal design strategy for a brushless AC (BLAC) motor. A schematic depicting the strategy is shown in Figure 1.

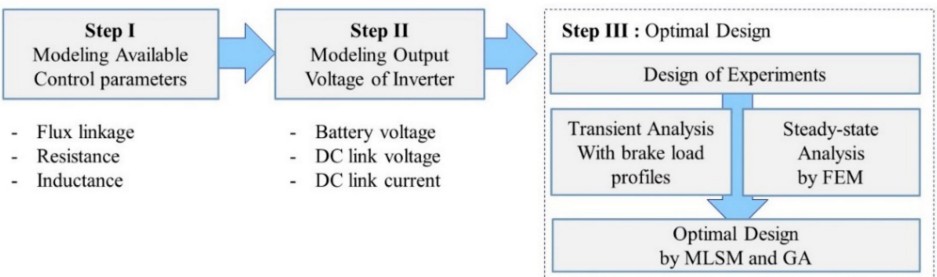

**Figure 1.** Optimal design strategy for the BLAC motor in an integrated brake system.

The strategy consists of three steps, as shown in Figure 1. In Step I, the motor control parameters are modeled for loading and ambient operating temperature conditions using the FEM. In Step II, the motor input voltage as an inverter output voltage, which depends on the loading condition and ambient operating temperature, is estimated by calculating the internal inverter voltage drop. Finally, in Step III, the optimal design is conducted using the modeled motor input voltage control parameters. In the optimization process, steady- and transient-state performances are set as objective functions, and the MLSM is used to implement the approximate modeling of the design variables and objective functions. In the LSM, if the response is complex, there is a limit to the global modeling; therefore, the MLSM was used [18]. Finally, the optimization results are obtained using the GA.

### 2.1. Modeling Motor Control Parameters

In Step I, the main control parameters and the motor input voltage are modeled as follows. As the main control parameters, the flux linkage, resistance, saturation coefficient, which is used to consider the magnetic saturation according to the current magnitude and phase angle, and motor input voltage, which varies according to the load conditions, are described sequentially. The flux linkage generated in the permanent magnet is proportional to the magnetic power density of the motor.

In addition, if the rated current is increased according to the load change, the magnetic flux density is saturated; consequently, the output density is reduced.

The resistance determines the copper loss and efficiency at the rated load. If the resistance increases according to the load change, the efficiency decreases; therefore, it must be considered when designing the motor.

### 2.1.1. Flux Linkage, Resistance, and Saturation Coefficient

The variation in flux linkage and resistance in response to different ambient operating temperatures is illustrated in Figure 2a. The variation in resistance with respect to the operating temperature is defined using the copper temperature coefficient, and the variation in the flux linkage is obtained through no-load FEM analysis. The saturation coefficient of the core is also distributed, as shown in Figure 2b, by considering the magnetic saturation effect.

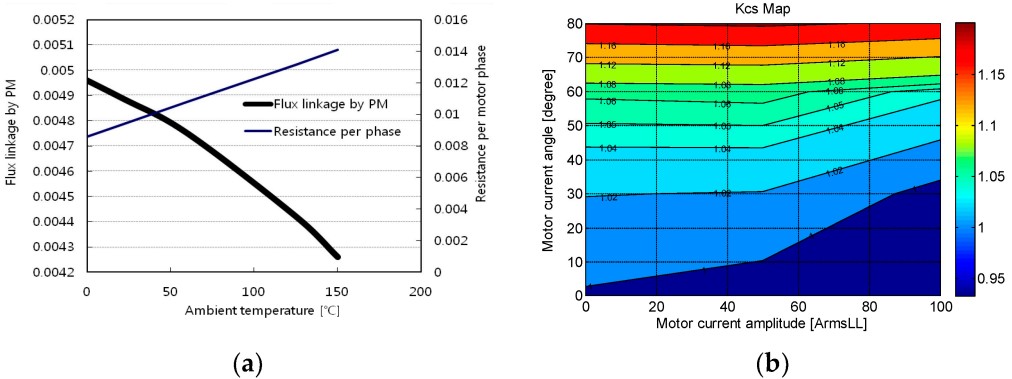

(a)                                        (b)

**Figure 2.** Calculation of Ψ, Rph, and Kcs: (**a**) flux linkage (Ψ) by PM and resistance (Rph) per phase; (**b**) map of the saturation coefficient (Kcs) by considering the magnetic saturation effect in the core.

### 2.1.2. d–q Axis Inductances

The d–q axis inductance is also an important control parameter when analyzing the motor performance. The inductance varies with the current and phase angle. In addition, an additional torque increase can be generated by the difference between the d-axis and q-axis inductances. This dependence of motor torque on the d and q-axis inductances can be used to control the motor speed. Figure 3 shows the inductance values of the d and q axes with respect to the magnitude of the motor current and phase angle.

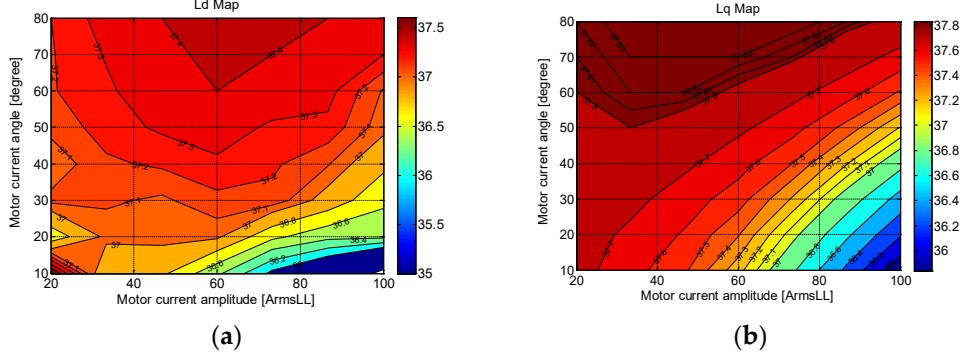

(a)                                        (b)

**Figure 3.** Calculation of the d- and q-axis inductance for a variation in the motor current amplitude and current angle: (**a**) Ld; (**b**) Lq.

### 2.2. Modeling of the Motor Input Voltage as an Inverter Output Voltage

In Step II, the motor input voltage is modeled as an inverter output voltage; this step consists of the following sub-steps:

1.   Simplify the resistances in the inverter system to Rdc and Rac;

2.    Input the load and calculate the base speed of the motor;
3.    Calculate the inverter output power;
4.    Calculate the input current of the inverter system at the current battery voltage; and
5.    Calculate the output voltage of the inverter system by subtracting the voltage drop across the inverter system from the battery voltage.

Figure 4 shows the process of estimating the motor input voltage as an inverter output voltage in detail. The internal voltage drop of the inverter can be calculated using Rdc, the total resistance connected in series with the DC-link input terminal of the inverter, and Rac, the total resistance connected to the inverter's AC output terminal. The voltage drop across Rac can be calculated by multiplying Rac by the inverter three-phase current measured using the shunt resistor connected to the three-phase output terminals of the inverter. In contrast, because there are no components, such as a shunt resistor, that can measure the current through Rdc, it is difficult to determine the voltage drop in Rdc. Therefore, the following process for calculating the inverter input current based on the motor parameters is introduced.

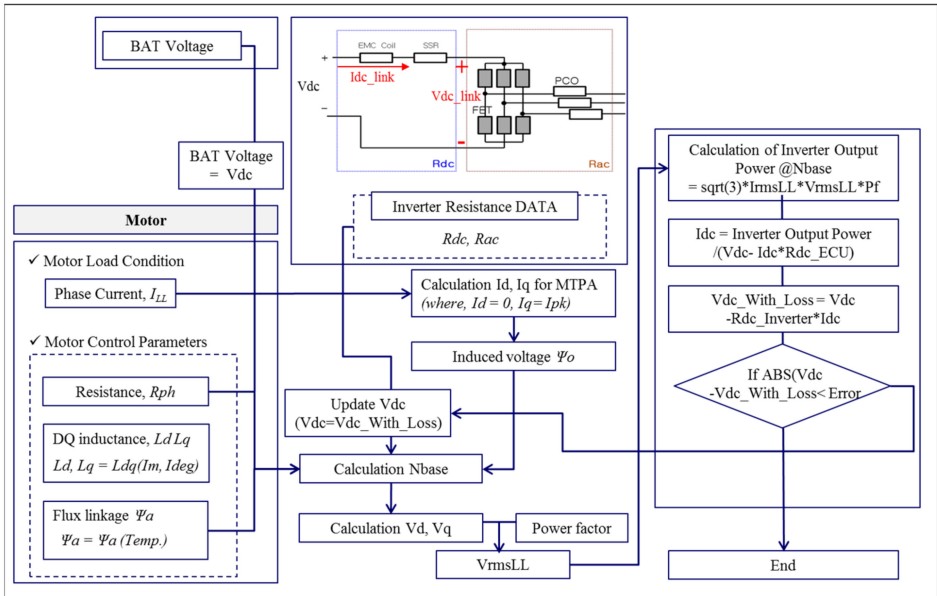

**Figure 4.** Calculation of the motor input voltage as an inverter output voltage.

In the first step, it is assumed that the DC-link voltage is equal to the battery voltage, and the inverter input power is calculated using the DC-link voltage and motor control parameters. In the second step, the inverter input current, that is, the DC-link current, is calculated by dividing the inverter input power by the DC-link voltage. The DC-link current is used to calculate the voltage drop across Rdc, and the DC-link voltage is updated. Based on the updated DC-link voltage, the parameter-based inverter input power and inverter input current are recalculated, and the DC-link voltage is repeatedly updated. When the inverter input battery voltage obtained through the above repetitive process converges to a definite value, the value is judged to be the battery voltage excluding the voltage drop across Rdc.

The performance of the motor is significantly affected by its input voltage, which is the same as the inverter output voltage. Because the output voltage of the inverter varies depending on the load condition and operating temperature of the motor, it is necessary to analyze it in consideration of this. For the variation in the ambient operating temperature and loading condition, the motor input voltage is estimated as the inverter output voltage as shown in Figure 5. The motor input voltage tends to decrease as the inverter output current and ambient temperature increase; this also increases the internal losses of the inverter.

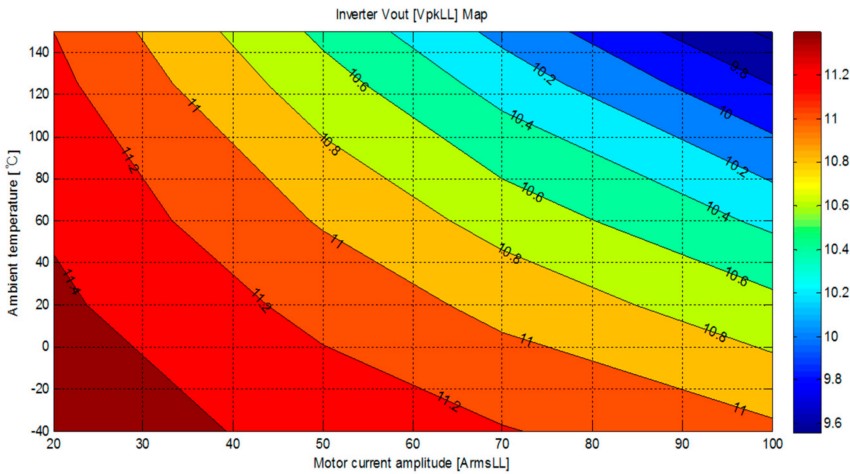

**Figure 5.** Map of the motor input voltage (inverter output voltage) for variation in the motor current amplitude and ambient temperature.

### 2.3. Optimal Design

The optimal design is conducted by utilizing the modeled motor input voltage and control parameters, considering the operating conditions, as shown in Figure 6.

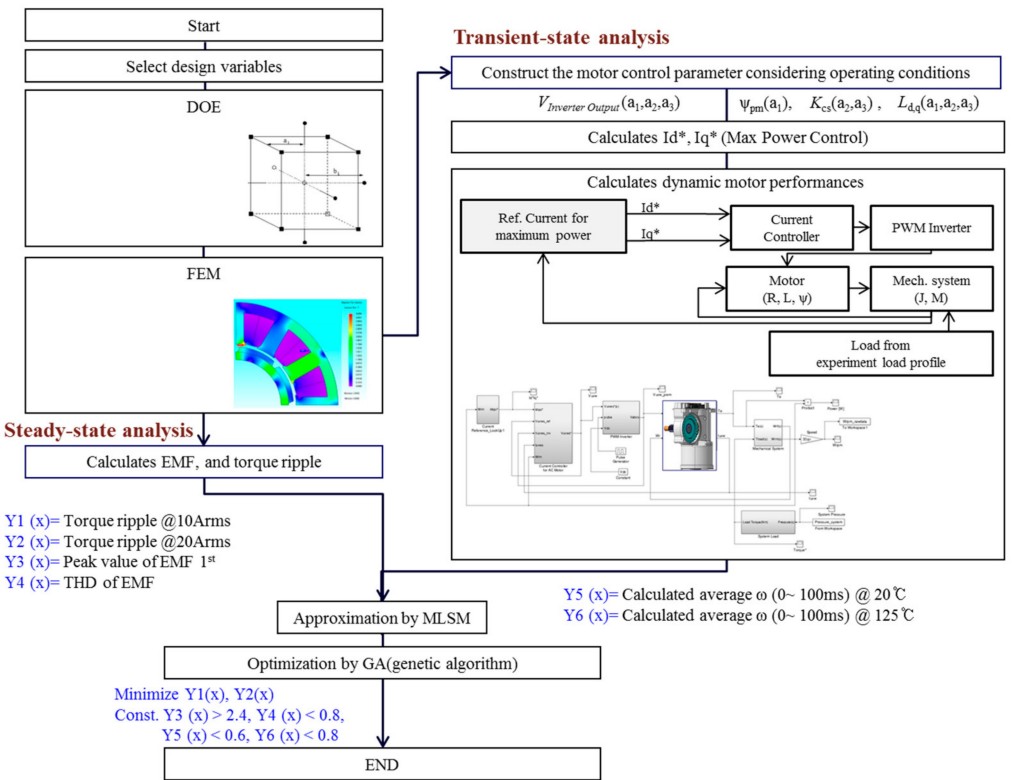

**Figure 6.** Optimal design process.

After analyzing the motor characteristics in the steady and transient states, the objective functions are approximated using the MLSM, and the optimal design is performed using the GA. In the steady state, torque-ripple and electromagnetic-force (EMF) characteristics are examined using the FEM; and, in the transient state, the speed response characteristics based on motor control parameter values are examined.

The important characteristics when designing an integrated brake system are the system noise and vibration, harshness (NVH), and rapid braking performance. In this

study, the motor performances corresponding to the system's NVH and rapid braking performance were selected as the objective functions. First, the torque ripple and total harmonic distortion (THD) of the motor's EMF were considered as the factors affecting the motor's performance corresponding to the system's NVH. Therefore, the torque ripple and THD of the EMF were selected as the NVH-related objective functions. Second, rapid braking performance is intricately related to motor control parameters and brake load conditions. Accordingly, Simulink was used to simulate the driving component of the integrated brake system, considering the motor control parameters and braking load conditions. The ratio of the target average speed of the motor to the average speed of the motor, which was calculated using Simulink under the braking load pressure profile, was selected as the objective function corresponding to the rapid braking performance. Because the rapid braking performance of a motor is affected by its operating temperature, the performance was analyzed at room temperature and at a high temperature. The selection of motor design variables in the optimal design process introduced in this study, transient analysis, DOE, and optimal design using the GA are explained sequentially.

### 2.3.1. Selection of Design Variables

The selected design variables are the outer radius of the permanent magnet (PM), the length of the slot opening, and the angle of the teeth shoe, which are denoted by x1, x2, and x3, respectively, as shown in Figure 7. The ranges of design variables were selected as listed in Table 1 by using central composite design (CCD) as the DOE [17].

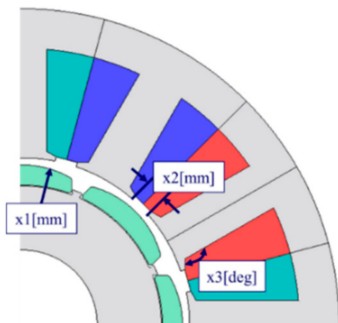

**Figure 7.** Selection of design variables.

**Table 1.** Selection of design variables with their ranges.

| Design Variables | Level of Design Variables | | | | |
|---|---|---|---|---|---|
| | −1.682 | −1 | 0 | 1 | 1.682 |
| x1 (mm) | 7.0 | 7.41 | 8.0 | 8.59 | 9.0 |
| x2 (mm) | 1.0 | 1.41 | 2.0 | 2.59 | 3.0 |
| x3 (degrees) | 125.0 | 129.05 | 135.0 | 140.95 | 145.0 |

### 2.3.2. Transient Analysis

Current vector control [16,20], considering the effects of electrical and mechanical time constants, was performed using the modeled voltage and motor control parameters, and the control strategy is illustrated in Figure 8. In this step, the experimental load profile was utilized.

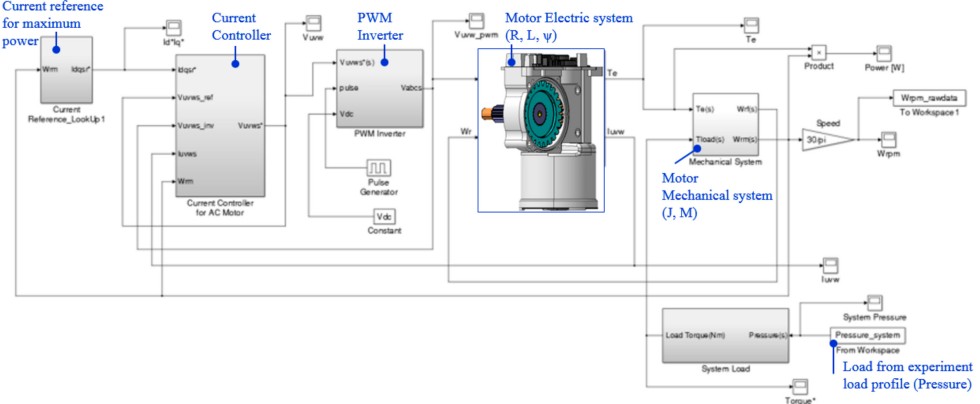

**Figure 8.** Current vector control strategy by using the experimental load profile.

The experimental load-pressure profile $P1_{LOAD}$ (*t*) in the rapid braking mode and the concept of the motor and pump gear system of the integrated brake system are illustrated in Figure 9.

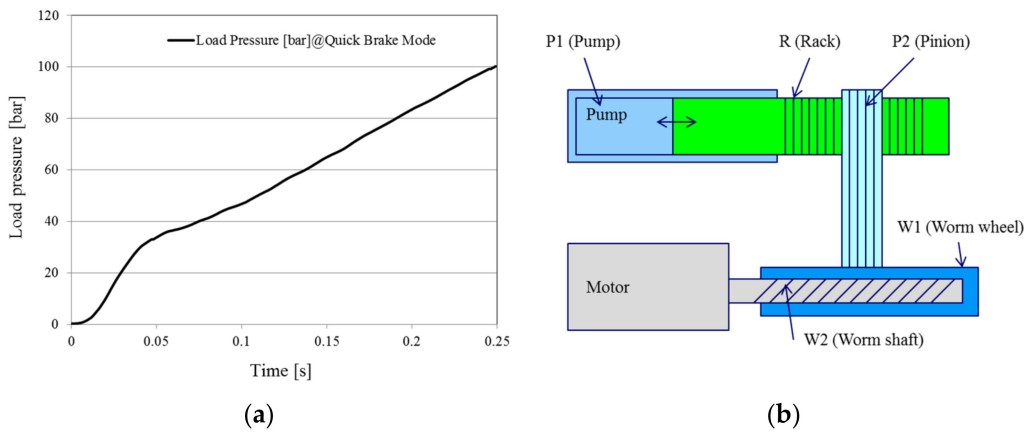

**Figure 9.** Load pressure: (**a**) experimental load pressure profile ($P1_{LOAD}$); (**b**) concept of the motor and pump gear system of the integrated brake system.

In the next step, the experimental load pressure profile $P1_{LOAD}$ (*t*) can be translated to load torque $T_{LOAD}$ (*t*) as

$$T_{LOAD}(t) = P1_{LOAD}(t) \cdot \left(\frac{R_{DIA}}{2}\right)^2 \cdot \pi \cdot \left(\frac{P2_{DIA}}{2}\right) / \left(\frac{P2_{EFF}}{100}\right) \cdot (W1_{GEAR_{RATIO}}) / \left(\frac{W1_{EFF}}{100}\right), \quad (1)$$

where $T_{LOAD}$ (*t*) and $P1_{LOAD}$ (*t*) are the load torque and load pressure profiles with respect to time obtained from the results of the experiments conducted on the integrated brake system, respectively; $R_{DIA}$ and $P2_{DIA}$ are the outer diameters of the rack and pinion gears, respectively; $P2_{EFF}$ and $W1_{EFF}$ are the efficiencies of the pinion and worm wheel gears, respectively; and $W1_{GEAR\_RATIO}$ is the worm wheel pinion gear ratio.

### 2.3.3. DOE with the Moving Least Square Method

These types of experimental designs are frequently used together with second-order response models. The design consists of three types of points, as shown in Figure 10.

- Axial points: $2 \times 3$ axial points are created by a screening analysis.
- Cube points: $2^3$ cube points are obtained from a full factorial design.
- Center point: A single point in the center is created by a nominal design.

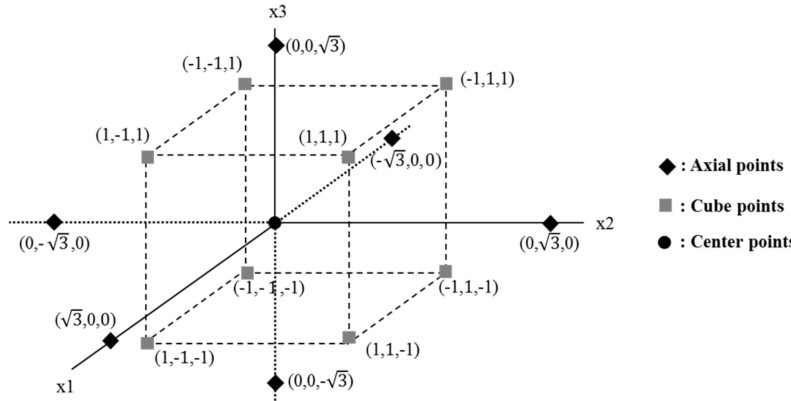

**Figure 10.** Example points of a central composite design with three input parameters.

The objective functions Y1–Y6 for the optimal design of the motor in the integrated brake system were constructed using the MLSM [18]. The following matrix expresses the relationship between the responses and the variables:

$$y' = \beta_0 + \sum_{i=1}^{k} \beta_i x_i + \sum_{i=1, j \leq i}^{k} \beta_i x_i x_j + \varepsilon = X\beta + \varepsilon, \tag{2}$$

where y' denotes the vector of the fitted values from the true function y, k is the number of design variables, X denotes the matrix of the levels of independent variables $x_i$ and $x_j$, β denotes the vector of the regression coefficients, and ε denotes the vector of random errors.

The least-square function Ly(x) is defined by the following equation, which expresses the sum of weighted errors as follows:

$$L_y(x) = \sum_{i=1}^{k} W(x)(y' - y)^2 = W(x)(X\beta(x) - y)^2, \tag{3}$$

where W(x) denotes a weighting matrix as a function of the location, which is obtained from the weighting function, y is a known value obtained from the simulation results, and N is the number of CCDs.

The weighting function W(x) uses the Gaussian function expressed in (4) and depicted in Figure 11.

$$w(x) = exp\left(-\frac{x^2}{h^2}\right) \tag{4}$$

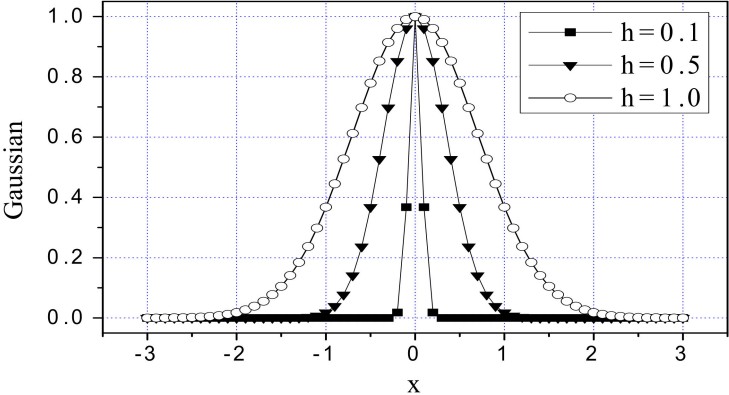

**Figure 11.** Gaussian weighting function.

The weighting matrix W(x) is constructed using the weighting function W(x) in the diagonal terms as follows:

$$W(x) = \begin{bmatrix} w(x - x_1) & 0 & \cdots & 0 \\ 0 & w(-x_1) & & 0 \\ & \vdots & \ddots & \vdots \\ 0 & 0 & \cdots & w(x - x_N) \end{bmatrix},$$ (5)

where x denotes the vector of the approximate location, and $x_i$ denotes the vector of the *i*th sampling point.

To minimize Ly(x), the least square estimators must satisfy the following expressions:

$$\frac{\partial L_y(x)}{\partial \beta(x)} = 2X^T(W(x)(X\beta(x) - y)) = 0$$ (6)

$$2X^T(W(x)(X\beta(x))) = 2X^T(W(x)y)$$ (7)

The coefficients of the regressive equation β(x) are obtained using the matrix operation as follows:

$$\beta(x) = \left[X^T W(x) X\right]^{-1} X^T W(x) y,$$ (8)

where the coefficient β(x) denotes a function of position x. It should be noted that the procedure for calculating β(x) denotes a local approximation, and the moving processes perform a global approximation throughout the design domain.

The simulation results of the CCD are listed in Table 2.

**Table 2.** Simulation results of the CCD.

| No. | x1 | x2 | x3 | Y1 | Y2 | Y3 | Y4 | Y5 | Y6 |
|-----|------|------|------|---------|--------|--------|--------|--------|--------|
| 1 | −1 | −1 | −1 | 2.7777 | 2.4891 | 2.4098 | 0.9818 | 0.5948 | 0.7532 |
| 2 | 1 | −1 | −1 | 2.7046 | 2.0805 | 2.5052 | 1.1708 | 0.5846 | 0.7288 |
| 3 | −1 | 1 | −1 | 7.2381 | 4.7215 | 2.345 | 1.2286 | 0.6255 | 0.6862 |
| 4 | 1 | 1 | −1 | 9.0465 | 5.3346 | 2.4386 | 1.3631 | 0.6615 | 0.6908 |
| 5 | −1 | −1 | 1 | 1.9623 | 1.2832 | 2.4112 | 0.7847 | 0.616 | 0.7592 |
| 6 | 1 | −1 | 1 | 2.3721 | 1.2798 | 2.5066 | 0.9943 | 0.5856 | 0.7396 |
| 7 | −1 | 1 | 1 | 6.1271 | 3.3318 | 2.3482 | 0.7633 | 0.616 | 0.6877 |
| 8 | 1 | 1 | 1 | 8.2165 | 4.3131 | 2.4422 | 0.9487 | 0.6492 | 0.6836 |
| 9 | −1.682 | 0 | 0 | 2.6175 | 1.9796 | 2.3434 | 0.7475 | 0.5846 | 0.7221 |
| 10 | 1.682 | 0 | 0 | 4.8375 | 2.6315 | 2.5066 | 1.089 | 0.6018 | 0.6907 |
| 11 | 0 | −1.682 | 0 | 1.3142 | 1.1542 | 2.4732 | 0.9063 | 0.6409 | 0.7813 |
| 12 | 0 | 1.682 | 0 | 10.3900 | 5.5766 | 2.3656 | 1.0736 | 0.6559 | 0.6909 |
| 13 | 0 | 0 | −1.682 | 6.5699 | 5.5594 | 2.4321 | 1.6452 | 0.6004 | 0.7005 |
| 14 | 0 | 0 | 1.682 | 4.8118 | 2.5999 | 2.4375 | 0.8809 | 0.5829 | 0.703 |
| 15 | 0 | 0 | 0 | 4.9424 | 2.8981 | 2.4367 | 0.9547 | 0.5921 | 0.7008 |

The objective functions were modeled using the MLSM, such as meta modeling [18]. Figure 12 shows the response surfaces of the objective functions according to the change in the design variables x1 and x2 at x3 = 0.

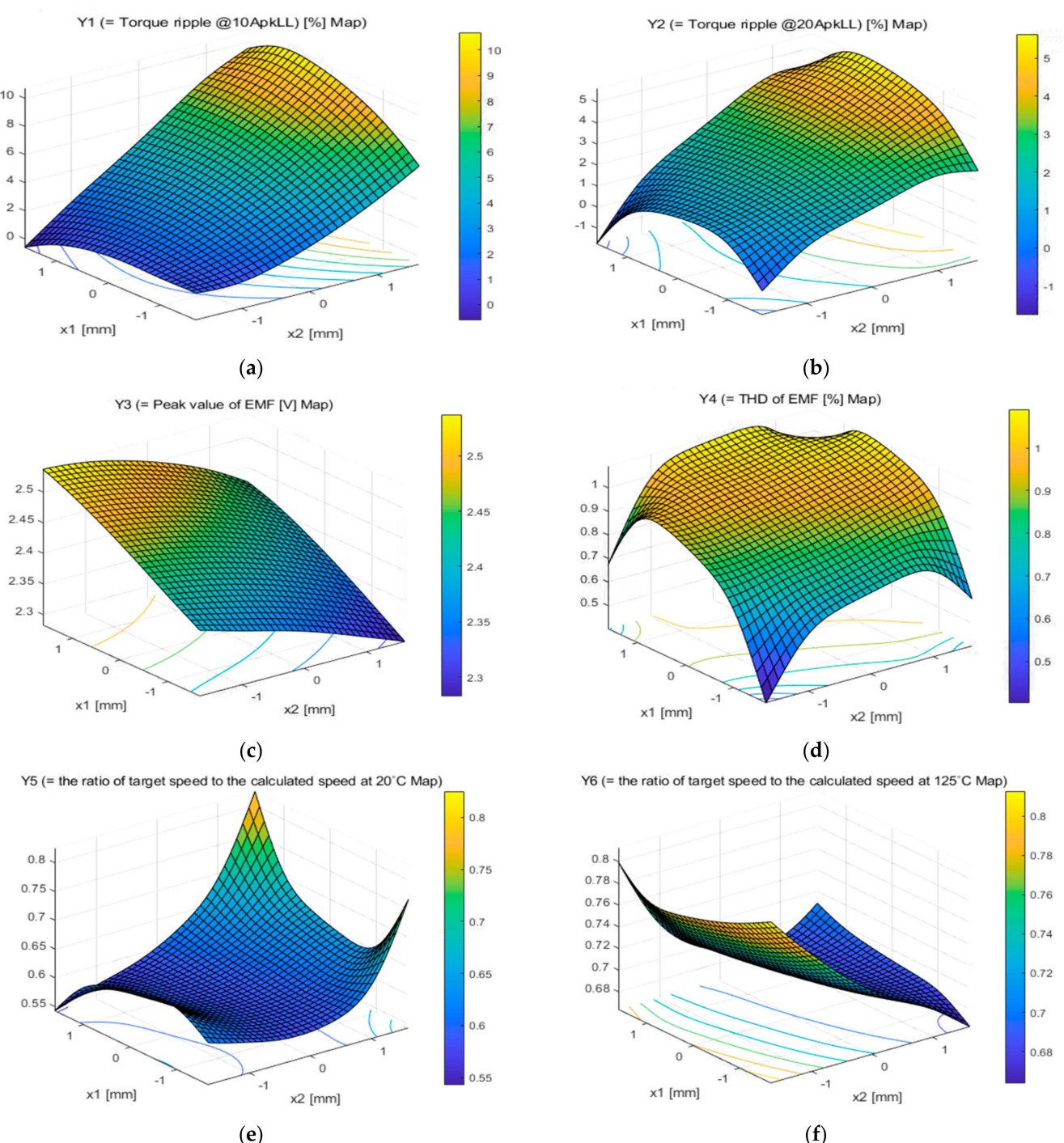

**Figure 12.** Response surfaces for objective functions at X3 = 0 (deg): (**a**) Y1; (**b**) Y2; (**c**) Y3; (**d**) Y4; (**e**) Y5; (**f**) Y6.

### 2.3.4. Optimization by the GA

The objective functions were constructed to minimize the torque ripple, and the constraint functions were selected as follows:

- Minimize Y1(x), Y2(x); and
- Constraint: Y3(x) > 2.36, Y4(x) < 0.8, Y5(x) < 0.6, Y6(x) < 0.8,

where Y1(x) is the torque ripple at 10 ArmsLL and 20 °C, Y2(x) is the torque ripple at 20 ArmsLL and 20 °C, Y3(x) is the peak value of the EMF at 20 °C, Y4(x) is the THD of the

EMF at 20 °C, and Y5(x) and Y6(x) are the ratios of the target speed to the calculated speed at 20 °C and 125 °C, respectively.

The process of searching for an optimal point for the design variables and objective functions is depicted in Figure 13.

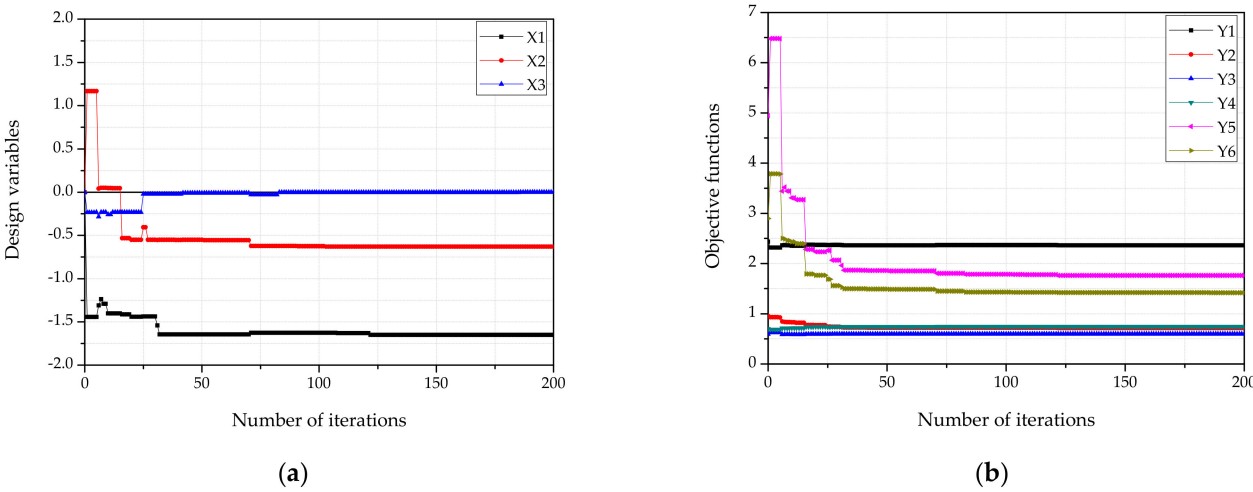

**(a)**                                                  **(b)**

**Figure 13.** Convergence history plot: (**a**) design variables; (**b**) object functions.

As a result of the optimal design, the design variables x1, x2, and x3 were calculated as 7.03 mm, 1.63 mm, and 135°, respectively. The approximate modeling was performed using the 15 motor design results listed in Table 2 and the MLSM, and the motor design variables satisfying the target motor performance were determined using the GA. The optimized values are listed in Table 3. A 2D FEM analysis was performed to validate the performance of the optimized motor, modeled using the MLSM. The significant agreement between the optimized results approximated by the MLSM and 2D FEM results validates the effectiveness of the proposed optimal design process.

**Table 3.** Optimal design results.

| Items | Torque Ripple @20 °C | | Back EMF (1st) (VrmsLL) | Back EMF THD (%) | Target Speed/Calculated Speed (0–100 ms) | |
|---|---|---|---|---|---|---|
| | @10 ArmsLL | @20 ArmsLL | | | @20 °C | @125 °C |
| Optimized model by the MLSM | 1.7352 | 1.4548 | 2.3648 | 0.7108 | 0.5884 | 0.7512 |
| 2D FEM (Verification) | 1.8782 | 1.5757 | 2.3644 | 0.7171 | 0.5979 | 0.7986 |
| Error | 8.24% | 8.3% | 0.02% | 0.89% | 1.60% | 6.30% |

The main motor performances of the optimization model over the entire operating range are as follows. The torque speed characteristics of the motor according to the temperature and voltage are shown in Figure 14.

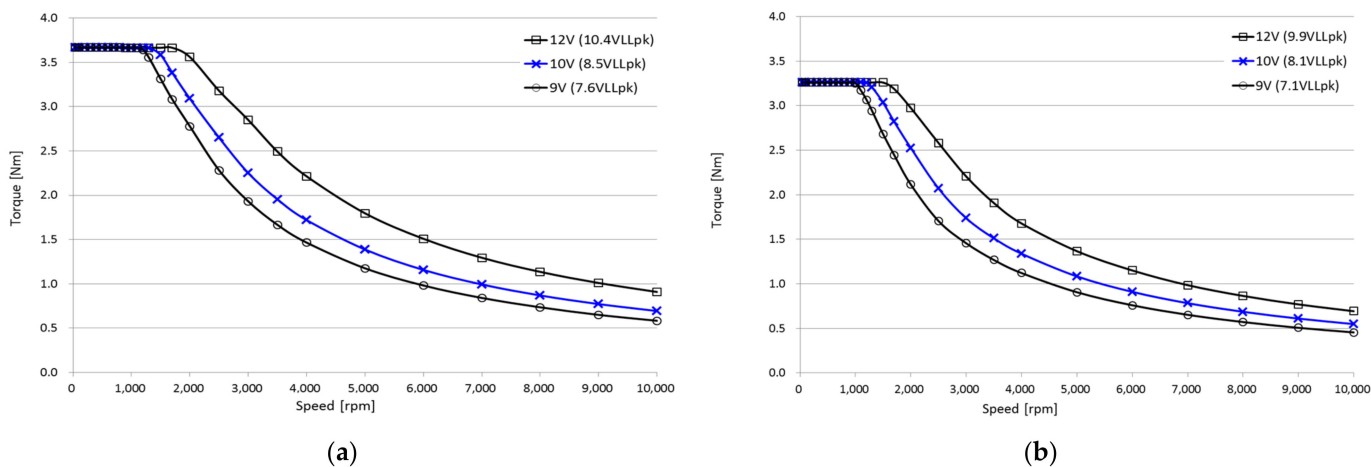

**Figure 14.** Torque speed characteristics: (**a**) 20 °C; (**b**) 125 °C.

Torque ripple and output power maps in the speed and torque regions at which the optimized model can operate are shown in Figure 15. In the main operation region of the motor, it has a torque ripple level of less than 2%, and it can be seen that it has a high output power characteristic of 800 W or more.

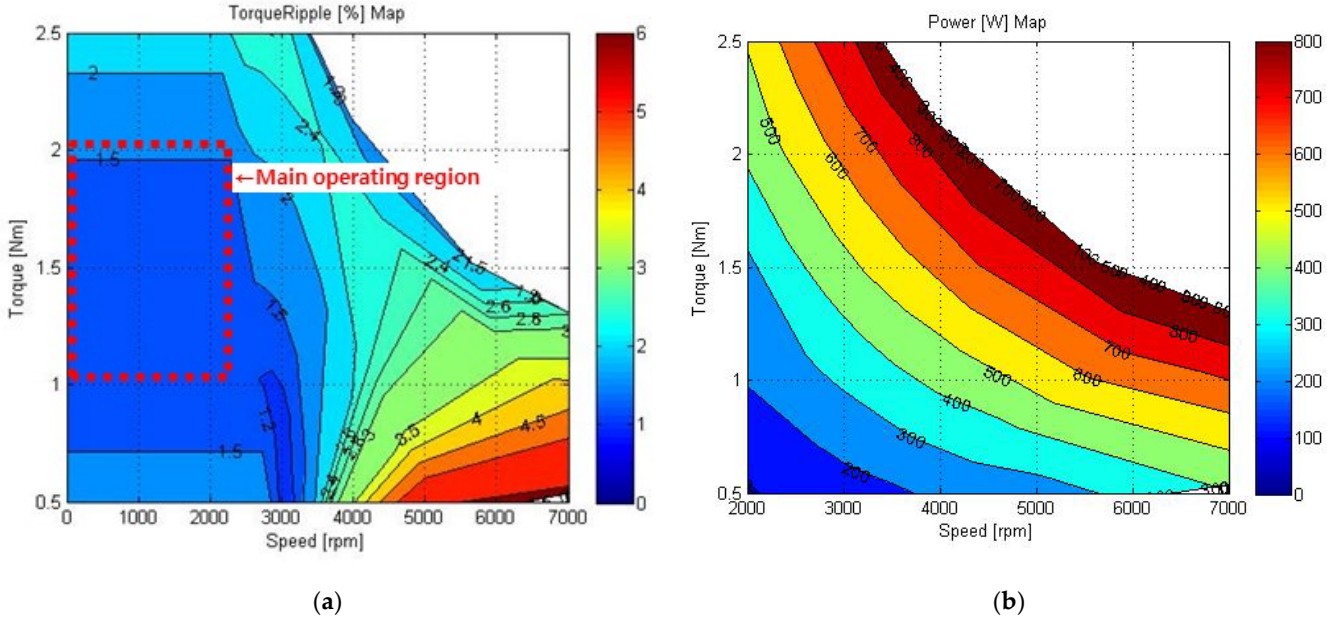

**Figure 15.** Motor characteristics maps of the optimized model: (**a**) torque ripple map; (**b**) power map.

The speed response of the optimized model during a rapid braking operation is shown in Figure 16a. As a result of the analysis, it can be confirmed that high-speed operation of up to 8000 RPM or higher is possible. Figure 16b shows the Id and Iq traces for the maximum power control during hard braking.

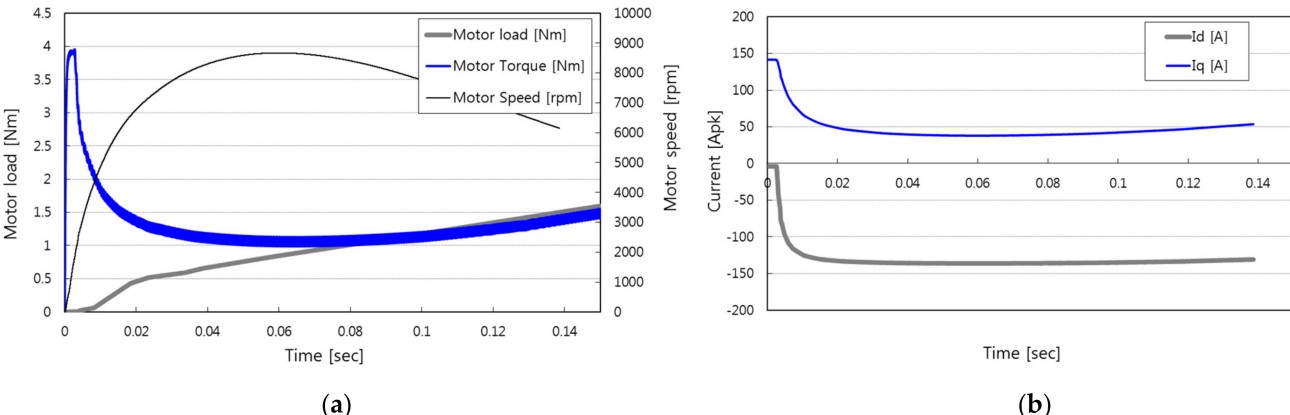

**Figure 16.** Transient performances of the optimized model in the rapid braking mode at 20 °C: (**a**) motor speed response; (**b**) Id and Iq.

### 3. Conclusions

This study investigated an optimal motor design process by considering variations in the motor control parameters and motor input voltages, considering the ambient operating temperature and loading conditions. Through the design process presented in this paper, it was possible to accurately consider the motor input voltages and control parameters, which are important factors when analyzing rapid braking performance in an integrated brake system. In addition, as a result of the optimal design, the performance of the motor in transient and steady states satisfied the requirements of an integrated brake system. The validity of the optimal design was confirmed through a comparison of the FEM analysis results with those of the proposed method. The experimental verification of the proposed method will be conducted as a future research project. Because the proposed design process can effectively consider the motor input voltage and parameters, which change according to the operating temperature and motor current, it can be applied not only to the integrated braking system but also to many other systems.

**Author Contributions:** Conceptualization, K.-Y.H.; methodology, K.-Y.Y.; software, K.-Y.H.; validation, K.-Y.H. and K.-Y.Y.; writing—original draft preparation, K.-Y.H.; writing—review and editing, K.-Y.H. and K.-Y.Y.; funding acquisition, K.-Y.Y. All authors have read and agreed to the published version of the manuscript.

**Funding:** This study was supported by a 2019 research fund from Honam University. This research was also supported by the "Regional Innovation Strategy (RIS)" through the National Research Foundation of Korea (NRF) funded by the Ministry of Education (MOE)(2021RIS-002).

**Institutional Review Board Statement:** Not applicable.

**Informed Consent Statement:** Not applicable.

**Data Availability Statement:** The data presented in this study are available on request from the corresponding author.

**Conflicts of Interest:** The authors declare no conflict of interest.

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
