# Peer review of "Optimal Design of an Inverter-Fed PMSM for a Brake System Considering Variation in Motor Control Parameters and Input Voltage"

_applsci, doi:10.3390/app12031707_

Round 1

Reviewer 1 Report

This paper continue your research in design optimization of the authors.  The introduction is concise and to the point.  The steps of the proposed designed are described clearly and I appreciate that you conducted the optimal design also by genetic algorithm. The transient state analyze is very important. The FEM analysis results confirm the validity of the proposed optimal design approach. In my opinion you made an important research. Congratulation!

Please revise English language (i.e the third phrase in the abstract is too long and not very clear. Less long propositions in the text, please).

Author Response

We greatly appreciate both your precious time and helpful comments.
The paper has been improved according to your suggestions and a response sheet was made as given below.

Reviewer 2 Report

You should compare the proposed esign approach to othe rexisting methods and show its superirity.

Author Response

(The authors gave the same response as above.)

Reviewer 3 Report

Study of optimal design of inverter-fed PMSM considering variation of motor control parameters and motor input voltage for integrated brake system is presented in the paper. The article is original, the topics are examined in a sufficiently consistent and detailed manner. The introduction provides an analysis of the literature that is quite brief. The authors could comment in more detail on individual sources, which are commented on very broadly in a larger group. The research part is arranged logically, providing a lot of visual material. The conclusions are based on the results of research.

In my opinion, the article does not have any major shortcomings and can be published. My recommendation to the authors is to expand the introductory part a bit by commenting in more detail on the individual sources.

Author Response

(The authors gave the same response as above.)

Reviewer 4 Report

In the reviewed paper, an optimal design process with consideration of the variation of motor control parameters and motor input voltage motor as an inverter output voltage have been presented. The paper is interesting but unfortunately underdeveloped. In my opinion, the paper can be published, after taking into account the following remarks:

  • the paper title is too long. The Authors should try to shorten it,
  • in the keywords section, the acronym "PMSM" should be written in full meaning, i.e. permanent magnet synchronous motor,
  • there is a lack of solid review of the scientific literature in the analyzed research field,
  • the article does not have the structure of a typical scientific article, consisting of an introduction (the current form is very short), a review of the literature on the subject, i.e. previous research by other authors in this field, a description of the materials and methods used, a description of the research, analysis, modeling, discussion over the results and conclusions obtained. It would be advisable to prepare the article in this form,
  • there is a lack of discussion and conclusion section,
  • the reference section is not formatted according to the Applied Sciences journal requirements.

Author Response

(The authors gave the same response as above.)

Round 2

Reviewer 2 Report

The paper has been improved according to my suggestions.

Good luck!

Reviewer 4 Report

The Authors have improved the paper according to the Reviewer's remarks.

Now, the paper can be published in a present form. Thank you very much.